# Genomic and Microscopic Analysis of Ballast Water in the Great Lakes Region

**David A. Wright** [1,2,*] **, Carys L. Mitchelmore** [1] **, Allen Place** [3] **, Ernest Williams** [3] **and Celia Orano-Dawson** [2]

[1] Chesapeake Biological Laboratory, University of Maryland Center for Environmental Science, Solomons, MD 20688, USA; mitchelm@umces.edu
[2] Environmental Research Services, Baltimore, MD 21209, USA; oranodawson@comcast.net
[3] Institute for Environmental Technology, University of Maryland Center for Environmental Science, Baltimore, MD 21202, USA; place@umces.edu (A.P.); williamse@umces.edu (E.W.)
*   Correspondence: david.wright105@comcast.net

**Abstract:** Invasive aquatic species can have damaging effects on fisheries and aquaculture through significant, and irreversible, effects on biodiversity. Human health may also be affected. To combat this threat the International Maritime Organization (IMO) Convention for Ballast Water and Sediments (BWMC) came into force in September 2017. U.S. Federal and IMO ballast water standards for discharged organisms stipulate discharge limits for different size classes of organisms. Several studies including recent trials aboard Great Lakes freighters have shown that many phytoplankton found in ballast water do not fall into the regulated 10–50 μM size class. Such issues illustrate the need for new methods of assessing microorganism populations that will supersede laborious microscopy requiring rare technical expertise. Recent progress has been made in the use of DNA (deoxyribose nucleic acid)-based methods as a means of identifying the appearance of invasive species in aquatic environments. A significant advance has been the development of high throughput sequencing (HTS), which has expanded DNA barcoding, relating to an individual organism, into second generation sequencing (metabarcoding), capable of mapping whole populations of organisms in an environmental sample. Several recent studies of HTS in ships' ballast water, have shown that the technique has the capacity for detecting potentially harmful taxonomic groups and is capable of differentiating among water from different sources. The current study was undertaken to investigate the suitability (or otherwise) of HTS as a tool for ballast water management. Possible applications include improved risk assessment relating to invasive species. Feasibility for indicative testing for ballast water treatment efficacy was also addressed. However, pending analysis of treated samples, the current study was confined to a comparison of HTS and microscope counts in untreated samples. A correlation of visual and molecular taxonomic assignments of microorganisms found in the ballast water from different ports and during different seasons indicated that such a comparison was best conducted at Family level, although Principal Components Analysis showed that the two methods differed qualitatively among major taxonomic groups.

**Keywords:** phytoplankton; great lakes; ballast water; nucleic acid; high throughput sequencing (HTS)

## 1. Introduction

Ships' ballast tanks may hold in excess of 100,000 tons of ballast water, transporting as many as 7000 species every day [1,2]. Invasive aquatic species can have damaging effects on fisheries and aquaculture through significant and irreversible effects on biodiversity. Human health may also be affected through the introduction of bacterial pathogens. Invasions have also resulted in huge

economic losses and costs. These range from medical costs to adverse effects on fisheries and recreation. They may also be responsible for fouling hard surfaces including industrial structures and water inlet pipes. Harmful algal blooms may result from the release of nonindigenous species into coastal waters, and releases of bacterial invasive species have been responsible for outbreaks of diseases such as cholera. In the U.S. the annual costs of bioinvasions has been estimated at over $100 billion [3] although perhaps one tenth of this figure relates to aquatic organisms. In the U.K. the annual cost of invasive species to the national exchequer has been estimated to be £1.7 billion [4,5]. To combat this threat the International Maritime Organization adopted the Convention for Ballast Water and Sediments (BWMC) in February 2004 [6]. This legislation sought to mitigate the risk of new invasions of non-native species through the implementation of standards governing the release of potentially harmful organisms during ballast water discharge. The interim D1 standard applied to mandatory ballast water exchange in ocean water, but following full ratification of the BWMC in 2017, this has been superseded by the D2 standard which stipulates post-treatment discharge limits for different classes of organisms, as follows [6]:

- Size class greater than 50 μM in minimum dimension (normally assumed to be zooplankton) Less than 10 viable organisms per cubic metre.
- Size class between 10 and 50 μM in minimum dimension. Less than 10 viable organisms per mL.
- Indicator bacteria:

  ○ *Vibrio cholera*; Less than 1 colony forming unit per 100 mL.
  ○ *Escherichia coli*; Less than 250 colony forming units per 100 mL.
  ○ Intestinal Enterococci; Less than 100 colony forming unit per 100 mL.

In addition to discharge densities for different classes of organism, shipboard certification tests also stipulate requirements for acceptable uptake (challenge) water densities, which are not always met. In a recent series of test of ballast water treatment aboard Great Lakes freighters [7] phytoplankton numbers in challenge water did not always reach the U.S. Federal (ETV 46, CFR 162.060-28) shipboard testing requirements for uptake densities. Levels of both indicator bacteria (*E. coli* and Enterococci) were also much lower than the prescribed uptake water levels for land-based testing. Additionally, the great majority of the phytoplankton identified in these trials appeared in the <10 μm size class, and as such were not subject to U.S. Federal or IMO ballast water regulation as standards are currently written [6]. A review of several other studies shows similar results. For example, Castro and Veldhuis [8] point out that many phytoplankton, including those forming harmful algal blooms such as *Phaeocystis* and *Pfiesteria* are smaller than 10 μM. Using size classification derived from flow cytometry, analysis of samples taken from the Wadden Sea showed that cells in the >10 μM size category represented only 3.6% of the total cell number and contributed only 28.7% of the total chlorophyll-a concentration. Other important dinoflagellates fall into the >50 μM, size class, normally associated with zooplankton, which are subject to a discharge standard $10^6$ times lower than the 10–50 μM size class. In waters around the British Isles Bradie et al. [9] reported that the >50 μM size class was dominated by *Ceratium* and *Protoperidinium*, which comprised 85% of individuals >50 μM. Therefore, strictly speaking these taxonomic groups would not be counted in the 10–50 μM (phytoplankton) regulated size class. In shipboard compliance testing off the Mexican coast, 12 out of 37 phytoplankton taxonomic groups (32.5%) were larger than 50 μM in the smallest dimension and 13/37 taxa (35.2%) were smaller than 10 μM in the smallest dimension. In terms of organism numbers, phytoplankton taxa <10 μM and >50 μM represented 14% and 27% of the total counts, i.e., only 59% of phytoplankton fell into the 10–50 μM size category [10]. Hess-Erga et al. [11] emphasize the omission of the <10 μM size category as an important deficiency in current discharge standards. Results such as these call into serious question the strict reliance on standards based on numbers of live or viable organisms within published size categories. Additionally, it has been noted that smaller (non-regulated) species may be faster growing and more easily adaptable to new environments [11–13]. This problem is recognized in the current development

of indicative tools for assessing compliance with the Ballast Water Convention. At the cellular level there is a need for new indicative tools for rapidly assessing the efficacy of ballast water treatment. Such tools, which may or may not be reliant on numeric standards include tagging with the vital stain Fluorescein Diacetate (FDA), chlorophyll fluorescence activity (CFA) and ATP determination using the luciferase enzyme. Indicators of cell viability may be partnered with flow cytometry to provide estimates of cell numbers. A recent, comprehensive shipboard study compared nine variants of these ballast water assessment technologies, five of which provided estimates of numbers of individuals [9].

Specific needs include the identification of 'high risk' organisms in ballast and receiving water and the development of rapid means of measuring the integrity of vital biological processes in treated organisms. Until recently nucleic acid sequencing had not been a focus for assessing cell viability in treated ballast water. While nucleic acid levels in aquatic samples have shown good correlation with phytoplankton cell count, the ability of DNA to persist outside live organisms can lead to overestimates of living material in discharged ballast water. Nevertheless, progress has been made in the use of DNA-based methods as a means of identifying the appearance of invasive species in aquatic environments [14] A significant advance in this respect has been the development of high throughput sequencing (HTS), which has expanded DNA barcoding, relating to an individual organism, into second generation sequencing (metabarcoding), capable of mapping whole populations of organisms in an environmental sample. The attractiveness of the technique is its potential for defining population shifts related to season and geography as well as subtle population changes related to invasive organisms. Several recent studies have advocated the use of molecular techniques in the analysis of ballast water [15–17]. Pagenkopp Lohan et al. [16] used nucleic acid sequencing to characterize protist communities in ballast water derived from the east, west and gulf coasts of the United States. In a recent application of this technique to ballast water analysis, Gerhard and Gunsch [17] compared bacterial biomarkers identified by HTS and Most Probable Number counts in ballast water discharged at four major international marine ports in 2015–2016.

The current study investigated the use of HTS to identify the effect of location and season on the phytoplankton and bacterial flora of ballast water encountered seasonally by Great Lakes freighters, together with the possibility of identifying potentially harmful taxonomic groups. It took the form of a side-by-side comparison of microbiota using conventional light microscopy and HTS. As such it was designed to provide preliminary data to determine whether this approach has potential for invasive species detection in ballast water or other aquatic environments. Significant correlation with existing methods of ballast water examination (e.g., microscopy) will determine whether HTS has potential as an indicative means of testing for the effectiveness of ballast water treatment. As a first step in this approach, the current study was limited to a feasibility exercise in methods comparison using untreated water.

## 2. Materials and Methods

This study comprised a year-long survey of samples taken by Great Lakes freighters as they transported iron ore pellets, taconite, from sources such as Duluth and Two Harbors on the north shore of Lake Superior to manufacturing destinations such a St Clair, Detroit, Indiana Harbor and Ashtabula in the southern Great Lakes. Data collected included overall numbers of phytoplankton, taxonomic identification of phytoplankton taxa (including size), and a molecular identification (and relative abundance estimate) of both phytoplankton and bacteria (including rare groups) via high throughput nucleic acid sequencing (HTS). These analyses were performed on split samples shared between two different laboratories for light microscopy and HTS respectively.

### 2.1. Collection and Shipping of Water Samples

Ship personnel were sent all sampling and shipping equipment with instructions. During ballasting (roughly half-way through the cargo unloading sequence) a water sample was collected by ship personnel opening up the valve that taps into a ballast line, allowing water to flush stagnant lines for

2–3 min. Two 1 L samples were then collected in sterile containers provided. Bottles were labeled with the date and location, packed into the foam cooler with 2–3 ice packs (previously frozen at −20 °C) and shipped overnight to the University of Maryland Center for Environmental Science, Chesapeake Biological Laboratory, Solomons, in Maryland. A total of 55 samples were processed in this way over a one-year period between 29 September 2016 and 29 September 2017.

## 2.2. Processing of Samples

Upon receipt of samples part of the sample was fixed in Lugol's solution, and the remainder was divided into two aliquots. One (1 L) was sent for cell counts and taxonomic identification of the phytoplankton under light microscopy. The second 1 L sample was pre-filtered through a 100 um mesh filter to remove zooplankton. Following this initial filtration, 500 mL of the sample was then filtered through a 48 mm GF/F membrane filter with a pore size of 0.7 um. The sample was stored in the dark at −20 °C until molecular analysis of phytoplankton. The other 500 mL was filtered through a 0.2 um filter and again dried and stored in the dark at −20 °C for molecular analysis of bacteria.

Sample examination comprised a two-tiered strategy using both standard microscopy analyses coupled with molecular analyses (based on the approach by Xiao et al. [18]). This dual approach provided:

- An assessment of the total numbers of phytoplankton,
- The taxonomic groupings of phytoplankton present and their numbers,
- Size of the specific algal species present,
- Molecular analyses to provide an estimate of relative abundance of phytoplankton and bacteria including rare species.

## 2.3. Light Microscopy

Light microscopy of low-temperature preserved samples was performed using a stereomicroscope, following procedures outlined by Elskus et al. [7]. Sub-samples were preserved in Lugol's Solution for intensive taxonomy and determination of cell sizes. Typically, counts were made of >200 squares of a 1000 square counting grid using a Sedgwick Rafter Cell.

## 2.4. High Throughput Sequencing (HTS) for Phytoplankton (and Bacterial) Species

Although light microscopy is a powerful tool to identify and size taxonomic groups of phytoplankton species, and remains the primary technique used in most quantitative studies of ballast water, the method is very time consuming, requires highly skilled taxonomists, and will not identify rare species as only small volumes of water can be analyzed even following preconcentration. Molecular techniques have the capability of determining the relative abundance of all species in the sample. High throughput screening (HTS) approaches have been successfully applied to assess microbial diversity and phytoplankton community structure. For this study 250 mL water samples were filtered through PCTE filters (0.2 μm and 1.0 μm), then placed in 0.9 mL of Longmire's buffer (0.1 M Tris, 0.1 M EDTA, 10 mM NaCl, 0.5% (*w/v*) SDS) and shipped O/N on ice to the University of Maryland Center for Environmental Science, Institute for Marine Environmental Technology where DNA was extracted from the filter and amplified with 16S and 18S primers for Illumina sequencing. Although the focus of this effort is to determine phytoplankton occurrence through taxonomic recognition of nucleic acid sequences, data was also obtained on the bacterial species present and their relative abundance in the sample.

Samples in DNA preservation solution were spun to collect material that had detached from the polycarbonate filter and the preservation solution was removed by aspiration. The lysis buffer from the Qiagen Power Soil DNA Extraction kit was added to each sample and any loose material was suspended by pipetting. The polycarbonate filter and the lysis buffer were then transferred back to the garnet bead containing tubes from the kit for lysis. Samples were split into subsets of

twelve for processing using the Qiacube with the Powersoil kit protocol and an additional inhibitor removal step. The resultant nucleic acids were quantified by absorption at 260 nm using a Nanodrop ND-1000 spectrophotometer.

The gene-specific sequences used in this protocol target the 16S V3 and V4 region followed the methods of Klindworth et al. [19]. The full-length primer sequences, using standard IUPAC nucleotide nomenclature, to follow the protocol targeting this region were:

16S Amplicon PCR Forward Primer = 5′
TCGTCGGCAGCGTCAGATGTGTATAAGAGACAGCCTACGGGNGGCWGCAG
16S Amplicon PCR Reverse Primer = 5′
GTCTCGTGGGCTCGGAGATGTGTATAAGAGACAGGACTACHVGGGTATCTAATCC

The algal specific primers were from the Bérard et al. [20]. vis.

18S Algal Amplicon PCR Forward Primer-5′
P73 (forward primer)—AAT CAG TTA TAG TTT ATT TGR TGG TACC
18S Algal Amplicon PCR Reverse Primer-5′
P47 (reverse primer)—TCT CAG GCT CCC TCT CCG GA

## 2.5. Determination of Biovolume

To express plankton community data within the framework of total biomass, it is necessary to estimate the relative contributions made to total community biomass by the taxonomic groups identified through nucleic acid sequencing. It has become common practice to estimate cell biovolumes through the application of geometric shapes and there are numerous literature sources recommending geometric shape assignment [20–26]. As detailed measurement required for the use of complex shapes is not part of the existing data, it was decided to base these calculations on the simplest geometric shape(s) possible using existing taxonomic data bases. It was decided that even a slight overestimation of biovolume was acceptable because taxa cell sizes vary greatly in estuarine systems due to environmental conditions. Overall dimensions have been compiled for 986 taxa found in the Chesapeake Bay Program Monitoring data. Currently there are 21 shapes employed in the standardized Chesapeake Bay List (Table 1). There are four notable simplifications to shape codes over Hillebrand et al. [23] recommended the shape. All taxa in the genera of *Amphora*, *Auricula*, *Cymbella*, *Encyonema*, *Hemidiscus*, *Rhopalodia*, where the recommended geometric shape is cymbelloid were simplified to ellipesoids. Shapes for the taxa of the genera *Actinastrum* and *Ankistrodesmus* were simplified to prolated sphere from cylinder+ 2 cones. Species in the genera of *Chataeteroceros* were estimated as cylinder, the recommended shape is elliptical prism but dimensions were unavailable to use the correct shape. Taxa in the genus *Chattonella* were estimated using a shape of a cone plus half sphere. The conversion of biovolume to carbon was based on protocol and equation from Smayda [27]. For all phytoplankton other than diatoms; the following equation was used to convert biovolume to a carbon estimate:

$$\text{Log}_{10}\ C = (0.886 \times \log_{10} (TV)) - 0.46$$

where:

TV = Total Cell Biovolume in cubic microns

C = Cell Carbon Value in pictogram per cell

For all diatoms, it is necessary to correct plasma volume for the cell vacuole, and then a conversion to carbon may be made. The cell cytoplasmic layer is calculated as the ratio of total cell surface area to area to total cell volume. (Table 1)

**Table 1.** Correction of Plasma Volume (PV) for Cell Vacuole.

| Ratio of Total Cell Area to Total Cell Volume | Cytoplasmic Layer Thickness |
| :---: | :---: |
| X < 0.35 | 2 μm |
| $0.35 \leq X > 0.50$ | 1.5 μm |
| 0.50–0.89 | 1 μm |
| >0.90 | PV = Total Volume |

Cell plasma volume was then calculated using an equation from Smayda [27].

$$\text{Plasma Volume} = (\text{Surface Area}, \mu M^2)(\text{Cytoplasmic Layer Thickness}) + (0.1 \times \text{Total Cell Volume}, \mu M^3)$$

Then the following equations are used to convert plasma volume to a carbon estimate:

Equation to calculate phytoplankton carbon for diatoms where the ratio of cell surface area to cell volume is <0.9:

$$\text{Log}_{10}\, C = 0.892 \times \log_{10}(PV) - 0.61$$

Equation to calculate phytoplankton carbon for diatoms where the ratio of cell surface area to cell volume is >0.9:

$$\text{Log}_{10}\, C = 0.758 \times \log_{10}(PV) - 0.422$$

where:

$$PV = \text{Total Cell Plasma in Cubic Microns}$$

$$TV = \text{Total Cell Biovolume in pictogram per cell}$$

$$C = \text{Cell Carbon Value in pictogram per cell}$$

This left approximately 405 taxa either without dimensions or having carbon values calculated but the dimensions necessary to make a carbon estimate independently of the data generating lab.

For 71 of these taxa, biovolumes and surface areas were derived by computing a mean of all carbon value for all the taxa in the genus, family or group. These taxa have assigned method designations of FAMILY_AVE, GENUS_AVE, or GROUP_AVE. For all the remaining 332 taxa, the provided carbon value was accepted as best available and a biovolume was back calculated using the appropriate equations from Smayda [27]. In all cases where diatom biovolumes were back calculated, the plasma volume was assumed to equal to Total Cell volume. (method codes: CBP_EST_T1 and CBP_EST_TB).

For a subset of the remaining 332 taxa where biovolume was back calculated, it was possible to derive an estimate of cell dimensions. Any cell where the assigned shape was a sphere or a cube requires only one dimension to calculate a biovolume. Resulting dimensions were spot checked against available taxonomic resources to assess whether derived values were realistic values. There were 96 taxa where both biovolume and dimensions were back calculated (method code: CBP_EST_TB). There were an additional 352 taxa where carbon values provided but the taxa are not currently found in the Chesapeake Bay Program monitoring data (method CBP_EST_T2). These data had their average biovolume calculated as described in Table 2.

**Table 2.** Geometric Equations for the Estimation of Cell Volumes and Surface Areas.

| Code | Shape | Volume | Surface Area |
|------|-------|--------|--------------|
| 1 | SPHERE | $\pi/6*D3$ | $\pi*D^2$ |
| 2 | CYLINDER | $\pi/4*D2*H$ | $\pi*D*(D/2 + H)$ |
| 3 | ELLIPSEOID | $\pi/6*D*H*W$ | $\frac{\pi}{4}*(D+W)*$ $\left[\left[\frac{D+W}{2}\right] + \frac{2H^2}{\sqrt{4*H^2-[D+W]^2}}\sin^{-1}\sqrt{\frac{4*H^2-[D+W]^2}{2D}}\right]$ |
| 4 | CONE | $\pi/12*D^2*H$ | $\frac{\pi}{2}*D*\left[\left[\frac{D}{2}\right] + \sqrt{H^2 + \left[\frac{D}{2}\right]^2}\right]$ |
| 5 | TRAPEZOIDAL PRISM | $1/2*H*DE*(W + W2)$ | SHAPE NOT IN CURRENT USE |
| 6 | CUBE | $W^3$ | $6*W^2$ |
| 7 | RETANGULAR BOX | $H*W*DE$ | $(2*H*W) + (2*W*DE) + (2**H*DE)$ |
| 8 | TRUNCATED CONE | $\pi/12*H*(D^2 + (D*D2) + D2^2$ | $\pi/4*(D2^2 + D^2 + 2H*(D2 + D)$ |
| 9 | TRIANGULAR PRISM | $1/2*H*W*D$ | $(W*DE) + (3*H*W)$ |
| 10 | ELLIPTICAL PRISM | $\pi/4*H*W*DE$ | $\pi/2*(H*W + (H + W)*DE$ |
| 11 | CONE-HALF SPHERE | $\pi/12*D^2*(H + D)$ | $1/2*\pi*D*(L + D)$ |
| 12 | NOT ASSIGNED | | |
| 13 | NOT ASSIGNED | | |
| 14 | DUMBELL | $2*(\pi/6*D^3)$ | $2*\pi*D^2$ |
| 15 | PRISM ON PARALLELOGRAM | $\pi/2*H*W*DE$ | $W*H + \frac{\sqrt{H^2+W^2}}{4}*DE$ |
| 16 | PYRAMID | $1/3H*W*DE$ | SHAPE NOT IN CURRENT USE |
| 17 | CYLINER-2 HALF SPHERES | $\pi*D^2*((H/4 + D/6)$ | $\pi*D*(D + H)$ |
| 18 | PROLATE SPHERE | $\pi/6*D^2*H$ | $\frac{\pi*D}{2}*\left(D + \frac{H^2}{\sqrt{H^2-D^2}}\right)*\sin^{-1}\left(\frac{\sqrt{H^2-D^2}}{H}\right)$ |
| 19 | 2 CONES | $\pi/12*D^2*H$ | $\pi*D*\sqrt{H^2 + \left(\frac{D}{2}\right)^2}$ |
| 20 | HALF SPHERE | $\pi/12*D^3$ | $3\pi/4*D^2$ |
| 21 | CYLINDER + CONE | $\left(\frac{\pi}{4}*D^2*H\right) + \left(\frac{\pi}{12}*D^2*H2\right)$ | $\left(\pi + \left(\frac{D}{2}\right)^2\right) + \left(2*\pi*\frac{D}{2}*H\right) + \left(\pi*\frac{D}{2}*\left(H^2 + \left(\frac{D}{2}\right)^2\right)\right)$ |
| 22 | CYMBELLOID | USE ELLIPESOID EQUATIONS | USE ELLIPESOID EQUATIONS |
| 23 | 2 ELLIPOSEOID | $\pi/6*D*H*W*2$ | $\frac{\pi}{4}*(D+W)*$ $\left[\left[\frac{D+W}{2}\right] + \frac{2H^2}{\sqrt{4*H^2-[D+W]^2}}\sin^{-1}\sqrt{\frac{4*H^2-[D+W]^2}{2D}}\right]^2$ |
| 24 | SICKLE-SHAPED PRISM | $\pi/4*H*DE*W$ | $\pi/4*(H + W + DE*W + H*DE) + H*DE$ |

## 2.6. Comparison of Taxonomic Groups Determined by Microscopy and Molecular Sequencing

Molecular taxonomic assignments were compared to the visual assignments using family level output of the Silva pipeline. Relative abundances within this taxonomic level were summed based on an approximation of the visual taxonomic assignment as follows: "Diatoms" summed from D_0__Eukaryota;D_1__SAR;D_2__Stramenopiles;D_3__Ochrophyta;D_4__Diatomea; "Green Algae" summed from D_0__Eukaryota;D_1__Archaeplastida;D_2__Chloroplastida; "Cyanobacteria" summed from D_0__Bacteria;D_1__Cyanobacteria; and "Dinoflagellates" summed from D_0__Eukaryota;D_1__SAR;D_2__Alveolata;D_3__Dinoflagellata. In addition, other algae in the 18S dataset were summed from the following taxonomic assignments: D_0__Eukaryota;D_1__Cryptophyceae; D_0__Eukaryota;D_1__Haptophyta; and D_0__Eukaryota;D_1__SAR;D_2__Stramenopiles;D_3__Ochrophyta; excluding Diatomea. To do a relative comparison of the 16S and 18S dataset the eukaryotic algae in the 16S dataset were summed from D_0__Bacteria;D_1__Cyanobacteria;D_2__Chloroplast; and used to normalize the relative abundances of the cyanobacteria to the total relative abundances of algae in the 18S dataset. The relative abundances of "Diatoms", "Green Algae", "Cyanobacteria" and "Dinoflagellates" were then calculated from these pooled abundances by determining the proportion of each group out of all algal assignments, i.e., green algae, cyanobacteria, dinoflagellates, cryptophytes, haptophytes, and ochrophytes. Relative abundances of visual taxonomic assignments were calculated from raw counts. The samples that were common to the visual assignments and molecular assignments were then used in a principal component analysis using a matrix of taxonomic groups and sample number for each method of taxonomic assignment and compared graphically.

## 3. Results

### 3.1. Microscopic Counts

Microscopic examination identified over 100 specific taxonomic phytoplankton groups from nine different harbors. The most frequently sampled harbors were Burns Harbor Michigan (15 sampling visits) and Indiana Harbor (12 sampling visits) while Essexville Michigan, Superior Wisconsin and Two Harbors Minnesota were each sampled just once. Phytoplankton counts as determined by light microscopy are recorded in Supplementary Materials A Tables S1–S7. Taxonomic groups identified in this manner included potentially harmful organisms such as the cyanobacterium Microcystis, which was detected in numbers exceeding 500 per mL in Monroe MI in September 2016. Other cyanobacteria detected in high numbers included *Coelosphaerium* (Indiana Harbor—132/mL; Detroit MI, 53/mL, October 2017), *Anabaenopsis* (Essexville MI—170, January 2017), *Aphamizimenon* (Essexville MI—616/mL, June 2017) and *Pseudanabaena* (Essexville MI 298/mL, June 2017; Monroe MI—134/mL, June 2017). Densities of the cyanobacterium *Aphanocaspa* in excess of 100/mL were found in Monroe MI, Indiana Harbor, Burns Harbor) between September and November 2016, and at Burns Harbor, St Clair, MI and Monroe MI in October 2017. High concentrations of the diatoms *Achnanthes*, *Asterionella*, *Diatoma*, *Fragillaria* and *Navicula* were seen in Burns Harbor MI in October and November 2016, and of the green algae Gonium (742/mL) and two species of Scenedesmus (collectively 392/mL) in Burns Harbor in October 2016, High concentrations of *Fragillaria* were also recorded in Indiana Harbor in September 2017 (206/mL, M/V *Burns Harbor*; 172/mL, M/V *American Integrity*) and in Burns Harbor in September 2017 (143/mL) and October 2017 (100/mL).The highest density of green alga was 946/mL recorded for *Aulocaseira* at Essexville MI in June 2017.

The relative contributions of different phyla to overall phytoplankton numbers are summarized in Table 3. At the Phylum level cyanobacteria identified by microscopy were generally dominant in warmer months as determined by overall relative densities of individual organisms per mL. However, the overall cell numbers and degree of dominance differed according to location. For example, very high numbers of cyanobacteria (500–4300/mL) resulted in 74–98% proportional dominance in Detroit in July–August 2017. However, cyanobacteria concentrations in Astabula, Ohio were low in August, but peaked in dominance in September/October 2017 (76.5%/71%).

Diatoms were proportionally dominant in Indiana Harbor, Indiana and nearby Burns Harbor in November/December 2016, and were also dominant in Monroe MI in September 2016, although no winter collections were conducted at the latter site. Further east, at St. Clair and Detroit, south of Lake Huron, diatoms dominated phytoplankton community numbers in May.

Green algae were locally dominant at Indiana Harbor in October 2016 (78–82%) although no collections were made from this location in October 2017. At Burns Harbor 46 Km to the east this group represented 22.6–41.8% and 15.5–44.4% of total phytoplankton numbers in October 2016 and October 2017 respectively. At other sites the proportion of phytoplankton represented by green algae varied between 0–59.3%, with little clear pattern related to season or location.

Dinoflagellates represented only a small proportion of the phytoplankton assemblage at all locations throughout the study period. The highest percentage reached by this group was 3.9% at Burns Harbor, Indiana in July 2017. However, at most locations and collection dates, dinoflagellates comprised less than 2% of overall phytoplankton densities.

**Table 3.** Summary of Major Phytoplankton Taxonomic Groups as Cell Nos. per mL of sample as a percentage total phytoplankton cells per sample.

**Indiana Harbor, Indiana**

| | 2 October 2016 | | 4 October 2016 | | 25 November 2016 | | 3 April 2017 | | 25 April 2017 | | 27 June 2017 | |
|---|---|---|---|---|---|---|---|---|---|---|---|---|
| | Cell No. | % | Cell No. | % | Cell No. | % | Cell No. | % | Cell No. | % | Cell No. | % |
| Diatoms | 109.3 | 5.1 | 28.3 | 1.9 | 219.3 | 78 | 36.3 | 18.8 | 109.6 | 77.2 | 0 | 0 |
| Green Algae | 1734 | 82.1 | 1188 | 78.8 | 27 | 9.6 | 34.5 | 17.9 | 17.6 | 12.4 | 46.2 | 61.8 |
| Cyanobact eria | 268.0 | 12.7 | 270.9 | 18 | 31.0 | 11 | 120.2 | 62.2 | 14.8 | 10.4 | 28.5 | 38.2 |
| Dinoflagell ates | 1.3 | 0.1 | 20 | 1.3 | 4.0 | 1.4 | 2.1 | 1.1 | 0 | 0 | 0 | 0 |

| | 27 July 2017 | | 22 August 17 | | 6 September 17 | | 13 September 17 | | 20 September 17 | | 25 September 17 | |
|---|---|---|---|---|---|---|---|---|---|---|---|---|
| | Cell No. | % | Cell No. | % | Cell No. | % | Cell No. | % | Cell No. | % | Cell No. | % |
| Diatoms | 28 | 7.7 | 200.1 | 24.6 | 187.3 | 5.5 | 325 | 41 | 225.7 | 21.3 | 42.2 | 1.8 |
| Green Algae | 88.0 | 24.2 | 109 | 13.4 | 288.2 | 8.4 | 162 | 20.4 | 74.3 | 7.7 | 138.0 | 5.9 |
| Cyano bacteria | 244.0 | 67.2 | 498.5 | 61.4 | 2926 | 85.8 | 304 | 38.4 | 750.0 | 71.4 | 2142 | 92.2 |
| Dino flagellates | 3.0 | 0.9 | 4.0 | 0.6 | 9.7 | 0.3 | 12.5 | 0.2 | 1.0 | 0 | 0 | 0 |

**Burns Harbor, Indiana**

| | 1 October 2016 | | 30 October 2016 | | 17 November 2016 | | 24 December 16 | | 3 December 16 | | 3 April 2017 | | 9 May 17 | |
|---|---|---|---|---|---|---|---|---|---|---|---|---|---|---|
| | Cell No. | % | Cell No. | % | Cell No. | % | Cell No. | % | Cell No. | % | Cell No. | % | Cell No. | % |
| Diatoms | 130.1 | 51 | 261.0 | 61.6 | 10423 | 78.6 | 341.4 | 80.9 | 96.2 | 83.1 | 2.8 | 13.4 | 857. 1 | 98.8 |
| Green Algae | 106.5 | 41.8 | 96.0 | 22.6 | 24.5 | 1.8 | 40.5 | 9.6 | 12.7 | 11 | 16.4 | 78.8 | 5.36 | 0.6 |
| Cyanobact eria | 11 | 4.3 | 65.5 | 15.4 | 256 | 19.3 | 36.0 | 8.5 | 6.4 | 5.5 | 1.2 | 5.8 | 0.9 | 0.1 |
| Dinoflagellates | 7.4 | 2.9 | 1.3 | 0.4 | 3.0 | 0.3 | 4.0 | 0.5 | 0.4 | 0.4 | 0.4 | 2 | 3.7 | 0.5 |

| | 17 May 17 | | 26 July 2017 | | 10 August 17 | | 13 September 17 | | 12 October 2017 | | 13 October 2017 | |
|---|---|---|---|---|---|---|---|---|---|---|---|---|---|
| | Cell No. | % | Cell No. | % | Cell No. | % | Cell No. | % | Cell No. | % | Cell No. | % |
| Diatoms | 100.4 | 79.3 | 220.5 | 49.3 | 294.0 | 81.5 | 326.5 | 29.9 | 120.0 | 40 | 297.0 | 41.3 |
| Green Algae | 2.1 | 1.7 | 142.0 | 31.8 | 55.0 | 15.2 | 69.5 | 6.3 | 133 | 44.4 | 111.5 | 15.5 |
| Cyanobact eria | 24 | 19 | 67.0 | 15 | 6.7 | 1.8 | 683.0 | 62.5 | 45.4 | 15.1 | 303.0 | 42.2 |
| Dinoflagellates | 0 | 0 | 17.0 | 3.9 | 5.0 | 1.5 | 14.2 | 2.3 | 1.0 | 0.5 | 7.0 | 1 |

**Monroe, Michigan**

| | 9 September 16 | | 5 October 2016 | | 13 October 2016 | | 28 September 16 | | 17 May 17 | | 23 June 2017 | |
|---|---|---|---|---|---|---|---|---|---|---|---|---|
| | Cell No. | % | Cell No. | % | Cell No. | % | Cell No. | % | Cell No. | % | Cell No. | % |
| Diatoms | 208.9 | 13 | 501.6 | 46.6 | 119.4 | 37.3 | 241.8 | 71.6 | 227.7 | 58.7 | 76.8 | 31.5 |
| Green Algae | 92.0 | 5.7 | 185.0 | 17.2 | 24.3 | 7.6 | 10.3 | 3 | 20.5 | 5.3 | 26.7 | 10.9 |

**Table 3.** *Cont.*

| | | | | | | | | | | | | |
|---|---|---|---|---|---|---|---|---|---|---|---|---|
| Cyanobacteria | 1066 | 66.5 | 292.0 | 27.2 | 176.5 | 55 | 85.7 | 25.4 | 139.6 | 36 | 76.0 | 31.2 |
| Dinoflagellates | 235.7 | 14.7 | 96.3 | 9 | 0 | 0 | 0 | 0 | 0 | 0 | 64.3 | 26.4 |

| St Clair, Michigan | | | | | | | | | |
|---|---|---|---|---|---|---|---|---|---|
| | 1 May 17 | | 14 May 17 | | 6 June 2017 | | 15 October 2017 | | 22 October 2017 | |
| | Cell No. | % | Cell No. | % | Cell No. | % | Cell No. | % | Cell No. | % |
| Diatoms | 212. 6 | 92.2 | 573.3 | 93 | 69.0 | 58 | 43.8 | 39.5 | 45.3 | 17.8 |
| Green Algae | 14.8 | 6.4 | 40.7 | 6.6 | 47.0 | 39.5 | 11.7 | 10.5 | 33.0 | 13 |
| Cyano bacteria | 3.2 | 1.4 | 0 | 0 | 0.7 | 0.8 | 54.4 | 49 | 175.4 | 69.2 |
| Dinoflagellates | 0 | 0 | 2.0 | 0.4 | 2.3 | 1.7 | 1.0 | 1 | 0 | 0 |

| Detroit, Michigan | | | | | | | | | |
|---|---|---|---|---|---|---|---|---|---|
| | 1 December 16 | | 17 May 17 | | 4 July 17 | | 7 August 17 | | 9 October 2017 | |
| | Cell No. | % | Cell No. | % | Cell No. | % | Cell No. | % | Cell No. | % |
| Diatoms | 55.7 | 72.2 | 500.0 | 98.6 | 99.0 | 14.6 | 66.5 | 1.5 | 82.0 | 29.6 |
| Green Algae | 1.8 | 2.3 | 3.0 | 0.6 | 77.0 | 11.3 | 17.0 | 0 | 54.0 | 19.5 |
| Cyano bacteria | 19.6 | 25.4 | 0 | 0 | 500.0 | 73.6 | 4299. 0 | 98.5 | 135.0 | 48.7 |
| Dinoflagellates | 0 | 0 | 4.0 | 0.8 | 3.0 | 0.5 | 3.0 | 0 | 6.3 | 2.2 |

| Ashtabula, Ohio | | | | | | | | |
|---|---|---|---|---|---|---|---|---|
| | 14 August 17 | | 30 August 17 | | 27 September 17 | | 2 October 2017 | |
| | Cell No. | % | Cell No. | % | Cell No. | % | Cell No. | % |
| Diatoms | 121. 1 | 12.8 | 75.2 | 33.7 | 570.2 | 13.5 | 166.0 | 15 |
| Green Algae | 70.8 | 7.4 | 132.5 | 59.3 | 394 | 9.3 | 151.5 | 13.7 |
| Cyanobacteria | 746 0 | 78.6 | 16.0 | 7 | 3235. 0 | 76.5 | 787.5 | 71 |
| Dinoflagellates | 11.3 | 1.2 | 0 | 0 | 30.0 | 0.7 | 3.0 | 0.3 |

| | ESSEXVILLE, MICHIGAN | | TWO HARBORS, MINNESOTA | | SUPERIOR, WISCONSIN | |
|---|---|---|---|---|---|---|
| | 7 June 2017 | | 23 October 2017 | | 31 October 2017 | |
| | Cell No. | % | Cell No. | % | Cell No. | % |
| Diatoms | 366 | 14.3 | 10.0 | 3.1 | 98.0 | 21.6 |
| Green Algae | 105 4.0 | 41.3 | 13.0 | 4.1 | 76.5 | 16.5 |
| Cyanobacteria | 110 8.0 | 43.4 | 237.0 | 92.8 | 284.5 | 61.4 |
| Dinoflagellates | 26.0 | 1 | 0 | 0 | 4.5 | 0.5 |

### 3.2. High Throughput Nucleic Acid Sequencing

The relative abundance of Operational Taxonomic Units (OTUs) was determined by High Throughput Sequencing (HTS), using both ends of DNA fragments in both reverse and forward orientation. No reference was made to microscopic data from shared samples in determining taxonomic grouping using HTS. Employing NGS Core Tools for longer reads and increased quality, a total of 26.5 million reads were reduced to 10.9 million usable sequences. Three poor quality samples (9, 12 and 21) were discarded after trimming and merging. The number of sequences examined per sample indicated that numbers of OTUs resolved stabilized at approximately 100,000 sequences per sample (Figure 1), where the accumulative differentiation of OTUs is expressed as rarefaction curves.

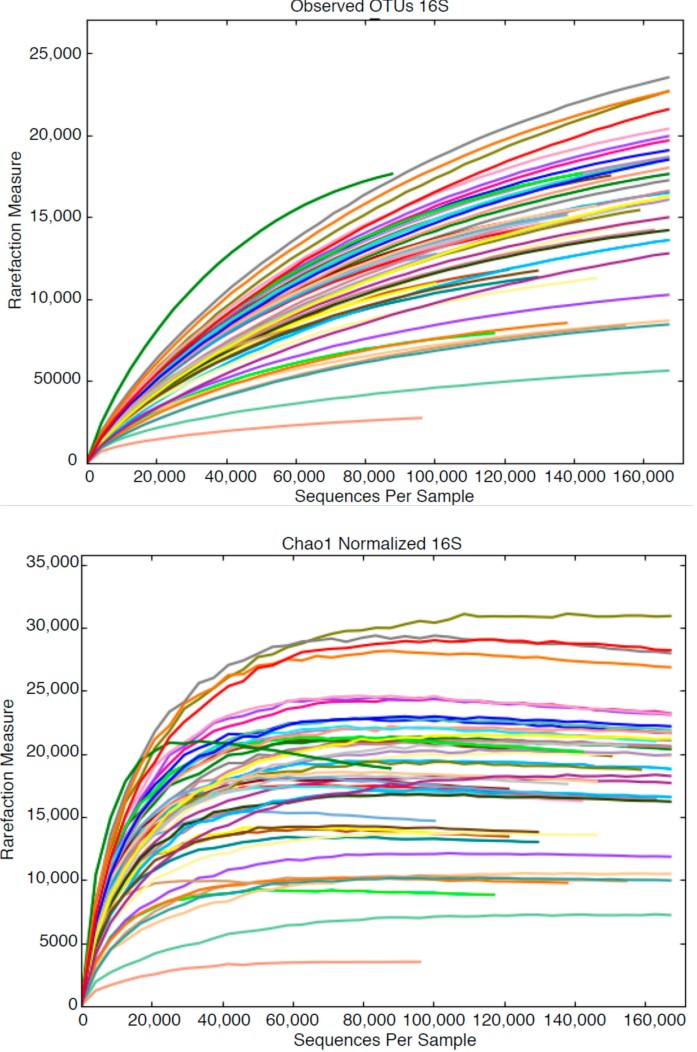

**Figure 1.** Rarefaction curves showing the relationship between sequences per sample and Operational Taxonomic Unit (OTU) resolution. In the lower graph, samples are categorized by lake.

Data indicated that additional sequencing beyond 100,000 will not provide greater coverage of the microbial assemblage. Sequencing data was edited for relevance to 18S. freshwater sample analysis. For example, major taxonomic groups identified were the algae Archaeplastida, Haptophyte, Cryptophyceae Archaeplastida, Haptophyte, Cryptophyceae and a super family of Stramenoplies. However, each sample contained protists such as *Centrohelida*, *Excavata* and Amoebozoa, and slime molds such as *Protosteliales*, which were irrelevant to 18S freshwater analysis and were not included in the analytical process. At 'Class' and 'Phylum' levels Stramenopiles, Archaeplastda and Chioroplasida were dominant groups, with Chlorophyta and Chlorophycae abundant at the 'Family' level, but with

few major differences between samples at this level. The data were too 'noisy' to interpret at the 'Genus' level. Using the QUIIIM script summarized_taxa_through_plots.py the summarized taxonomy was categorized according to three factors; sample, location and temperature.

Detailed representations of alpha diversity determined through HTS and related to location and temperature are shown in Supplementary Materials B. Supplementary Materials B Figure S1 shows alpha diversity for all samples at the 'Phylum' level, and in Supplementary Materials B, Figures S2 and S3 alpha diversity at the 'phylum' level has been categorized according to temperature and lake respectively. Alpha diversity is extended to 'Class' level in Supplementary Materials B Figure S4 and categorized according to temperature and lake in Supplementary Materials B, Figures S5 and S6 respectively. For analytical purposes temperatures were divided into three groups: 4–10 °C, low; 10–20 °C, medium and 20–26.8 °C, high. At the 'Phylum' level, the low temperature samples contained relatively higher percentage of Archaeplastida. Interestingly, the abundance of Cryptophyceae and Haptophyta was higher in the low temperature samples. A similar phenomenon was also evident at the 'Class' level, where there was a higher abundance of Archaeplastida and Chloroplastida in samples at medium and high temperatures. Higher proportions of Cryptophyceae_ Cryptomonadales and Haptophyta_Prymnesiophyceae were found in low temperature samples. The same trend was seen at the 'Order' level where the percentage of Archaeplastida_Chloroplastida_Chlorophyta was higher in the medium and high temperature samples. Chlorophyta_Sphaeropleales and Chlorophyta_Chlorophyceae were in relatively low abundance in the low temperature samples. At the 'Genus' level, almost all of the OTUs were not assigned, suggesting that 'Genus' level data were inappropriate for this analysis. Overall, the low temperature environment exhibited different features compared to higher temperature environments, indicating a critical role in determining the differential abundance of phytoplankton taxa.

Summarized taxonomy were also categorized according to the harbor where the samples were collected. At 'Phylum' level, except for samples from Two Harbors, major taxonomic group was Archaeplastida. In Two Harbors samples, the percentage of Cryptophyceae was higher than other location samples. Additionally, there were few dominant of protists and slime mold. At the 'Family' levels, there was a high and constant abundance of Archaeplastida/Chioroplasida/Chlorophyta/Chlorophyceae except for the Two Harbors samples. However, almost all OTUs were not assigned at both Family and Genus level. This suggests that it would be difficult to use the summarized taxonomy in these levels for the analysis. Overall, while summarized taxonomy by individual sample shows no major differences between the samples, there were small differences in the relative abundance of taxonomy in the Two Harbors sample. This suggests that Two Harbors has some specific environmental factors compared to other harbors. However, as there was only one sample from Two Harbors, further study would be necessary to confirm this.

In Figure 2 results of a Principal Components Analysis are expressed in graphical form showing molecular data groups 16S and 18S as a function of Harbor vs Season. Ellipses represent the 95% confidence limits for each sampling season. If a harbor is outside the other ellipses, it is statistically different at the 5% level.

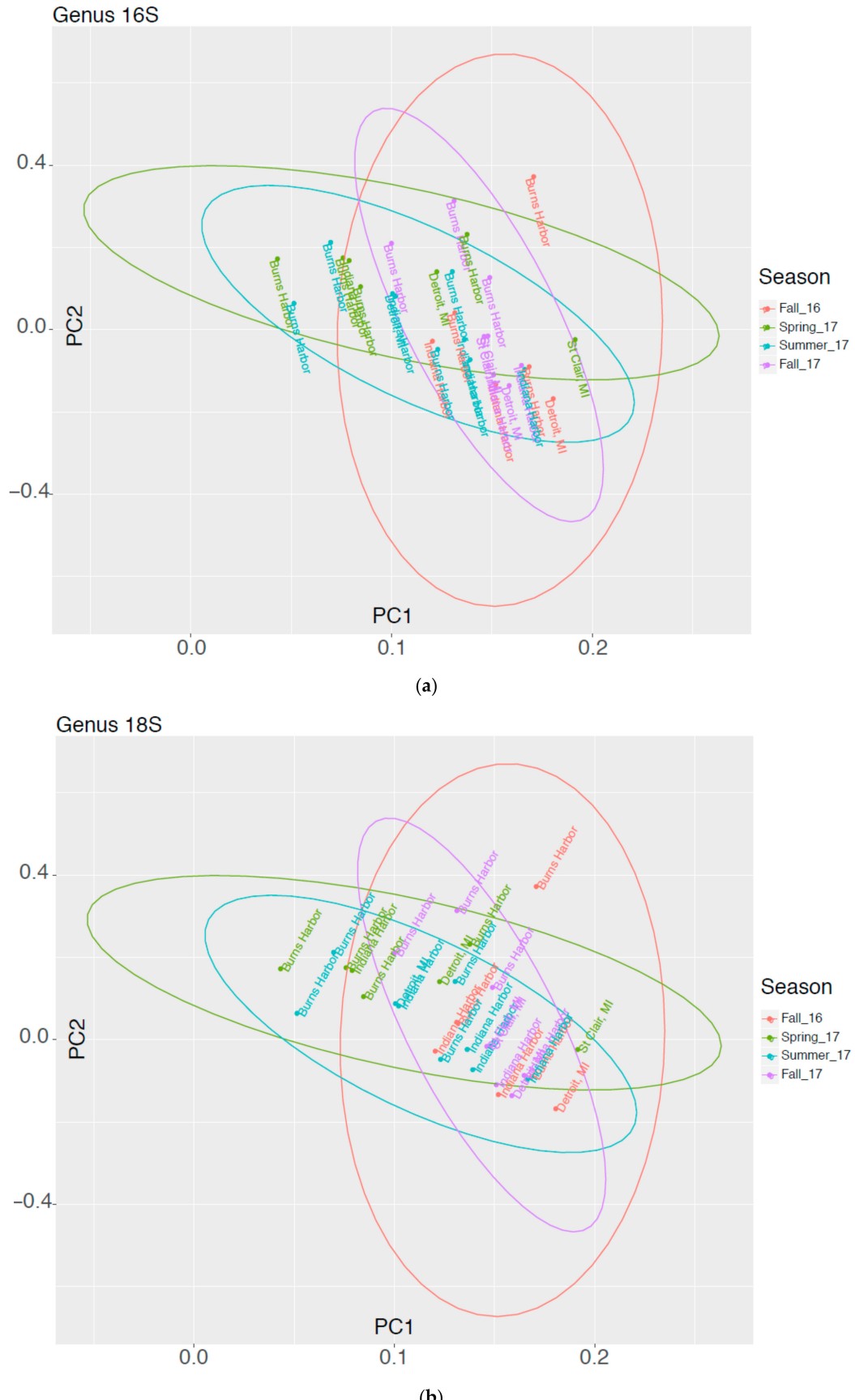

**Figure 2.** Graphical Representation of Molecular Data as a Function of Harbor and Season for (**a**) 16S and (**b**) 18S organisms.

*3.3. Comparison of Taxonomic Profiles Derived from Microscopy and Molecular Methods*

Forty-one Samples across nine geographic locations were compared for the relative abundance of "Diatoms", "Green Algae", "Cyanobacteria", and "Dinoflagellates" as identified in Table 2 using visual and molecular methods and assigned at the Family level according to the scheme described in Methodology Section 2.6. The relative abundance of dinoflagellates was consistently low across all samples using both methods (maximum in visual versus molecular assignment 8.9% and 0.08%, respectively). Green algae and cyanobacteria were generally present across all samples and methods (average of green algae and cyanobacteria 19.3% and 34.5% by visual assignment and 46.5% and 38.8% by molecular assignment, respectively), although the actual proportion and the dominant group commonly differed between methods. There were several samples where cyanobacteria were not observed by visual methods but had a nominal abundance according to molecular methods. The observation of diatoms was very inconsistent between microscopy and molecular determination, with very low abundance in all samples using molecular methods (maximum 0.03%) and low to very high abundances using visual methods (maximum 98.85%). Alpha diversity at the Family level is shown in Figure 3 (OTU 16S) and Figure 4 (OTU 18S) respectively, and principal component analysis of relative taxonomic abundance comparing the visual and molecular methods are shown in Figure 5.

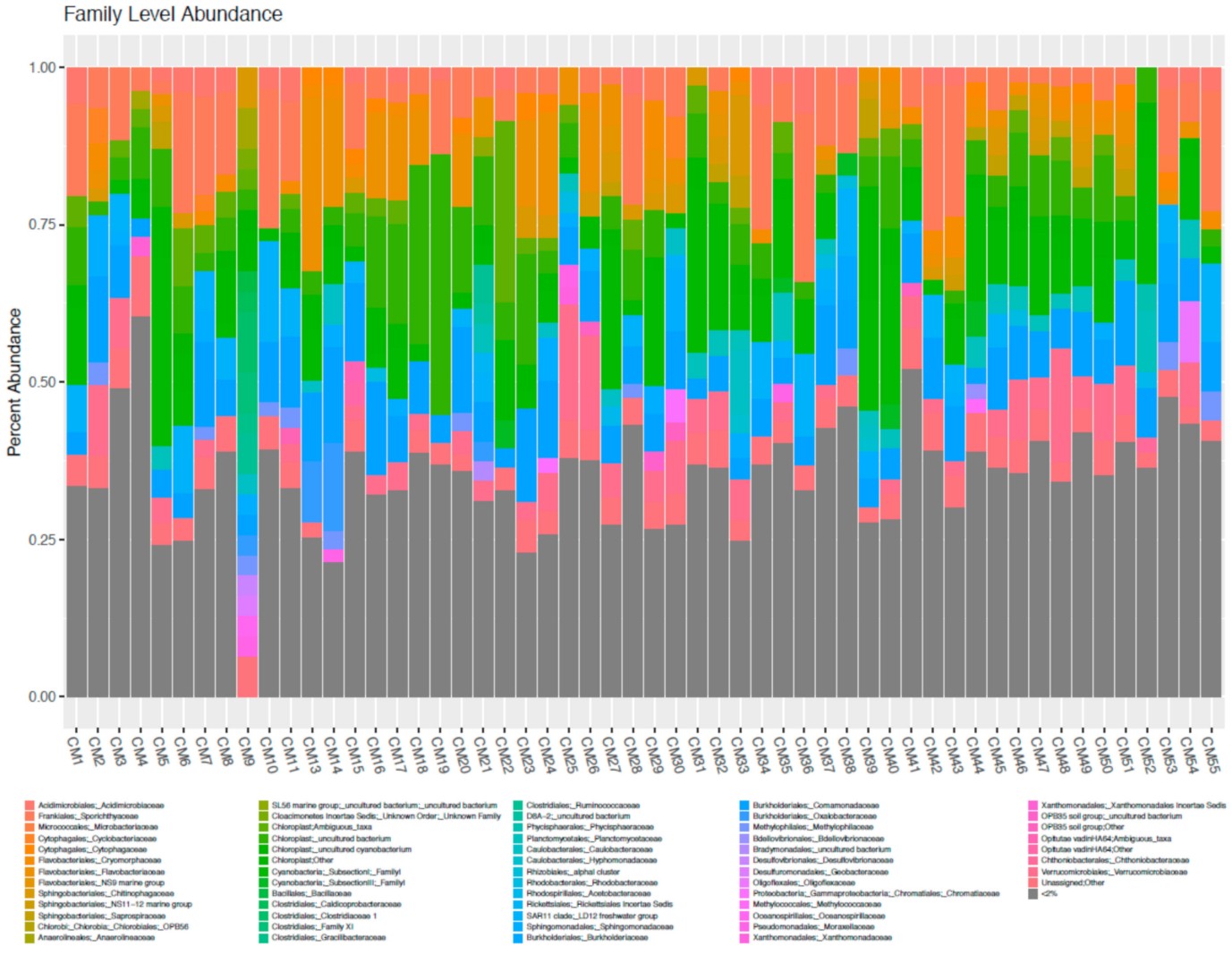

**Figure 3.** Alpha diversity at the Family level for OTU 16S.

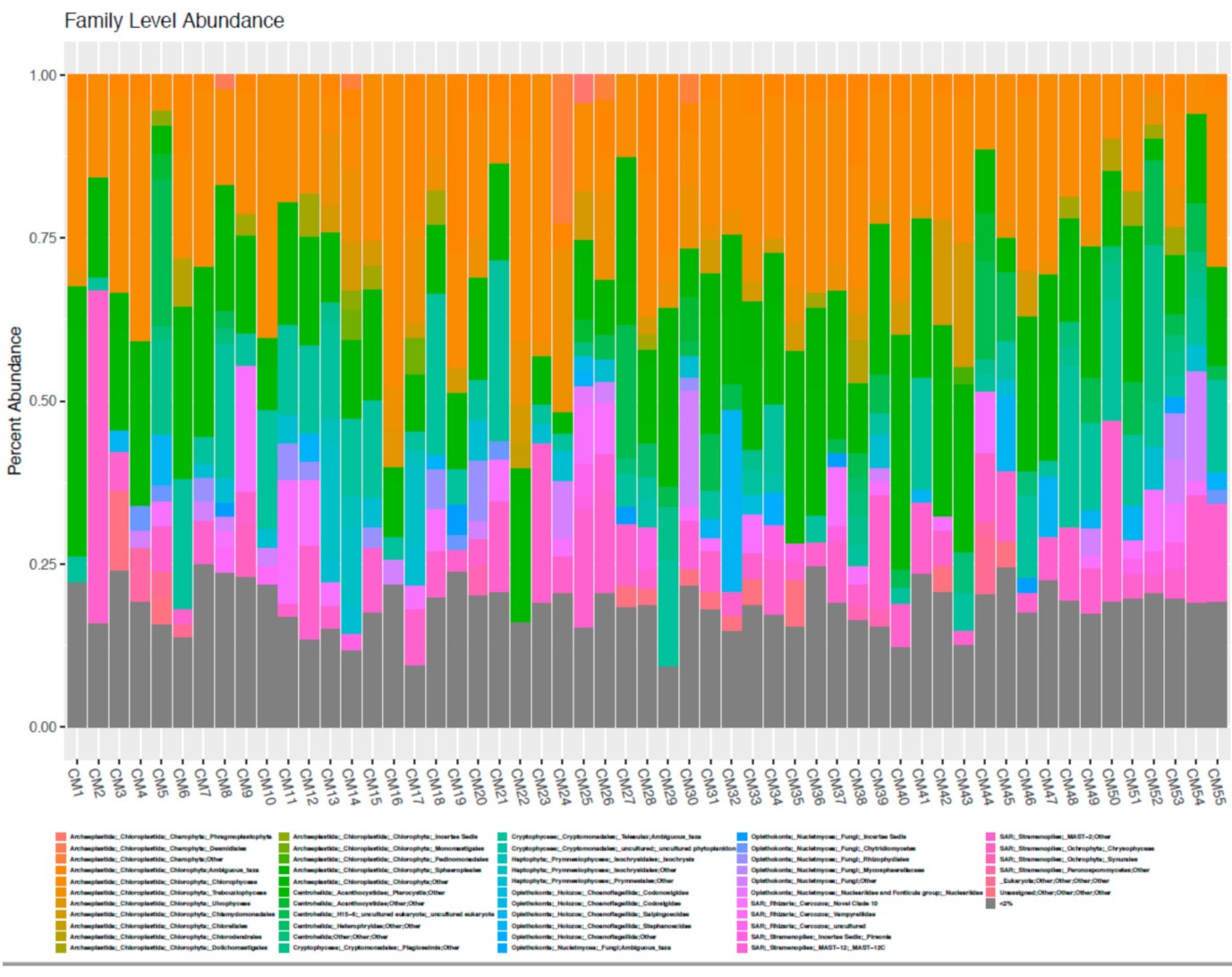

**Figure 4.** Alpha diversity at the Family level for OTU 18S.

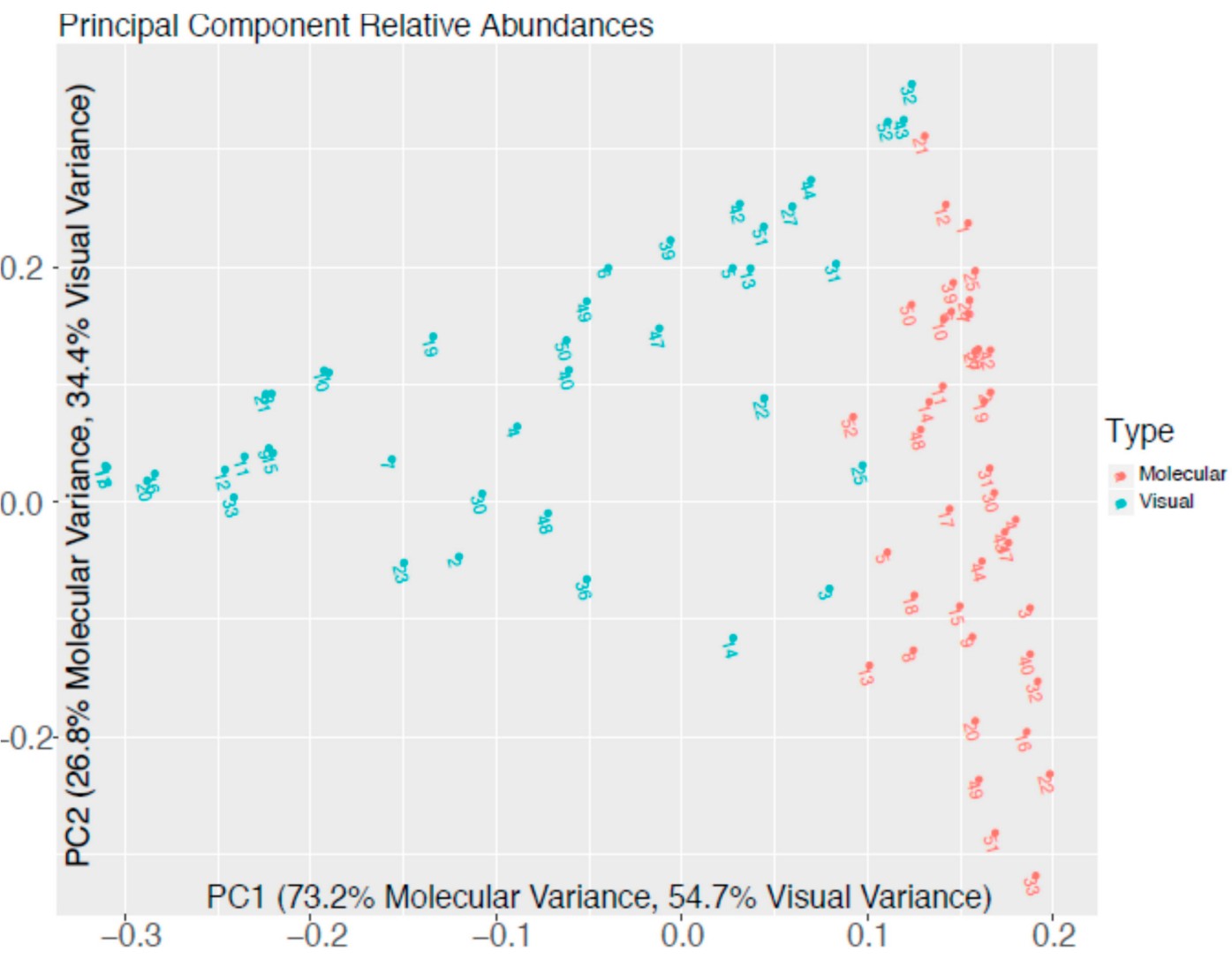

**Figure 5.** Principal Component Analysis of Relative Taxonomic Abundance Comparing Visual and Molecular Methods.

## 4. Discussion

While earlier shipboard tests of ballast water treatment [7] demonstrated the need for a seasonal testing regime, there is also a clear requirement for information on the seasonal and temporal variability of the phytoplankton flora in the Great Lakes, and the effect of seasonal and temporal changes on the size/taxonomic profile of these organisms. From the perspective of ballast water management, questions include how different locations and seasonal changes in taxonomic groups may affect the size profile of the phytoplankton fraction, with consequent changes to the challenge water characteristics. For example, a seasonal bloom of potentially harmful dinoflagellates or bacteria may provide a much greater challenge to treatment than at other times. Therefore, there is clear need for as much information as possible on the make-up of both bacterial and phytoplankton flora likely to be encountered by Great Lakes vessels plying their trade over a large spatial and temporal range. Microscopic phytoplankton results reported here show the numbers and types of phytoplankton that freighters will encounter in a normal year-long season in all of the locations that the ships visit. In a parallel study of harbor water and ballast water from coastal China released in the ports of Shanghai, China; Durban, S. Africa; Los Angeles, CA, U.S.A. and Singapore. HTS was used to identify bacterial biomarkers primarily used to assess water of origin [17]. Whole organism (i.e., non-molecular) counts differed from the current study in that, unlike the microscopic phytoplankton counts performed here, IDEXX Colilert® and Enterolert® MPN assays were used for counts of indicator bacteria [17]. An HTS analysis of ballast water arriving at east coast, west coast and Gulf coast ports in the U.S. was able to differentiate among protists arriving at different ports according to source water with specific reference to toxic dinoflagellates [16]. Although all of these studies clearly illustrate the potential for HTS to characterize the biome according to season and origin, the value of this technique as a tool for ballast water management is less clear.

While a major goal of this project was to examine the potential for nucleic acid sequencing to identify key taxonomic groups in ballast water from different locations and seasons, at least qualitatively, the potential for this technique as an indicative test for the effectiveness of ballast water treatment technology remains speculative. It must be emphasized that the results reported from this and other molecular surveys of ballast water were from untreated water. As such they are have the potential for tracking the presence and movement of potentially harmful species. However, in order to demonstrate the feasibility of this approach as an indicative tool for ballast water management it would be necessary to identify characteristic nucleic acid profiles for key taxonomic groups, and to demonstrate changes in nucleic acid sequences that correlate with diminished phytoplankton viability and/or reproductive potential following ballast water treatment. Nucleic acid sequencing has yet to be coupled with demonstration of ballast water treatment, and would require field trials of treatment including a comparison of nucleic acid sequencing with current compliance assessment methods and other indicative technologies currently in development [28]. HTS has the potential to provide a comprehensive picture of the planktonic flora at any particular time or location without resorting to highly specialized microscopy.

Comparison with conventional microscopy can provide important information on the utility of molecular techniques in identifying potentially harmful taxonomic groups in freshwater and marine environments. In the context of identifying potentially harmful invasive organisms, however, it is clear that substantial further refinement would be needed for molecular techniques such as HTS to be universally applicable. Clear differences between microscopy and HTS remain. For example, while potentially harmful cyanobacteria show reasonable equivalence between visual and molecular methods the same cannot be said of diatoms and dinoflagellates, where large discrepancies remain. Although internal consistency is generally high using visual and molecular methods for taxonomic assignment, the results of the two methods differ greatly from each other (Figure 5). There are several explanations for this that are inherent in the two methods. The likelihood of successful preservation is important for visual assignment of taxonomy with diatoms being generally easy to preserve and observe. Haptophytes, cryptophytes, and cyanobacteria can be more difficult to preserve and may be underrepresented using visual methods. Conversely, cryptophytes may be overrepresented using

molecular methods as environmental DNA resulting from frequent blooming can contaminate the sample and inflate estimates of extant members of the population. Likewise, high gene copy number in an organism can create errors in molecular estimates and are especially problematic with dinoflagellates given their very large and duplicated genomes [29]. This may explain why molecular estimates were consistently one or two orders of magnitude lower than visual estimates of dinoflagellates in this study. Another issue with molecular assignment is the sequence itself. Some taxonomic clades (descendants of a common ancestor) are difficult to sequence even with degenerate primers due to large numbers of pairwise differences between their sequence and other clades. Additionally, certain clades can be poorly represented in taxonomic databases and a conflation of these two issues can cause large underestimation of abundance. This is a likely explanation for the low abundances of diatoms observed by molecular methods. Combined with the ease of preservation and identification of this group of algae, this results in a large disparity in observed abundances using visual and molecular identification. This disparity is likely the dominant driver of differences observed between the methods (Figure 5), as samples with high abundances of diatoms had strongly negative eigen values on principal component axis 1. Likewise, relative abundances of green algae and cyanobacteria are likely drivers of axis 2 given the separation of the molecular assignments that are consistently low in the abundance of diatoms and dinoflagellates on this axis.

In view of discrepancies in the size classes of organisms regulated under the Ballast Water Convention [6] and several reports showing that large numbers of planktonic organisms often fall outside regulated size categories [7–11], there remains a clear need for indicative measures of ballast water treatment that are independent of organism size and do not require a high degree of taxonomic expertise. While HTS can provide important information on the sources and transport of potentially harmful organisms that could form the basis for risk assessment, its use as a regulatory tool appears limited. Such an application would require the development of a reliable molecular 'signature' based on the genetic integrity of samples following ballast water treatment. While the results reported here relate only to untreated ballast water, follow-up testing on treated water calibrated against microscopy would be needed to identify changes in nucleic acid profiles that reflect the elimination of harmful organisms. Disparity between HTS and microscopic counts of some microorganisms, as seen here, suggests that this approach may be limited in scope to specific taxonomic groups showing good correlation between microscopy and molecular profile. Current results indicate that cyanobacteria might fit this profile, although diatoms and dinoflagellates would not. Additionally, an important characteristic of a prognostic test for ballast water treatment efficacy is that sampling and analysis should ideally be completed during the period of a port visit. While the detailed taxonomic knowledge required for microscopy is not needed for HTS, the methodology for such molecular sequencing currently appears too lengthy to fulfil this time requirement. As such HTS is probably best suited to a broader risk assessment role, with prospects for further refinement.

**Supplementary Materials:** The following are available online at http://www.mdpi.com/2076-3417/9/12/2441/s1, Figure S1: Alpha Diversity at Phylum Level. [Sample Number (1–55)]. Figure S2: Alpha Diversity Categorized by Water Temperature at Phylum Level. Figure S3: Alpha Diversity categorized by Lake at Phylum Level. Figure S4: Alpha Diversity at Class Level. Figure S5: Alpha Diversity Categorized by Water Temperature at Class Level. Figure S6: Diversity Categorized by Lake at Class Level. Table S1: Microscopic Phytoplankton Counts in Ballast Water from Indiana Harbor, Indiana. Table S2: Microscopic Phytoplankton Counts in Ballast Water from Burns Harbor, Indiana. Table S3: Microscopic Phytoplankton Counts in Ballast Water from Monroe, Michigan. Table S4: Microscopic Phytoplankton Counts in Ballast Water from St. Clair, Michigan. Table S5: Microscopic Phytoplankton Counts in Ballast Water from Detroit, Michigan. Table S6: Microscopic Phytoplankton Counts from Ashtabula, Ohio. Table S7: Microscopic Phytoplankton Counts in Ballast Water from Essexville Michigan, Two Harbors Minnesota and Superior Wisconsin.

**Author Contributions:** The specific roles of the authors of this paper were as follows: Conceptualization, D.A.W., C.L.M. and A.P.; D.A.W. secured funding and logistical support from ASC; D.A.W. and C.L.M. were responsible for Project Administration; Liaison with ASC and Sample Processing were carried out by C.L.M.; Light microscopy was performed by C.O.-D. (ERS); A.P. and E.W. were responsible for nucleic acid sequencing and statistical treatment and presentation of molecular data; D.A.W. prepared the first draft of the manuscript, which was then reviewed and edited by C.L.M., E.W. and A.P.

**Funding:** This research was funded by a private grant from the American Steamship Company (ASC) to Environmental Research Services (ERS) who sub-contracted the University of Maryland Center for Environmental Science Chesapeake Biological Laboratory (CBL) and the Institute for Environmental Technology (IET) to perform molecular analyses. ASC played no part in sample processing and analysis or in the presentation or interpretation of results.

**Acknowledgments:** The authors are grateful for the extraordinary efforts to develop innovative approaches to ballast water treatment and testing by the American Steamship Company (ASC) as shown by their support of this work and previous studies. This project could not have been carried out without the initiative and support of Noel Bassett and Pierre Pelletreau of ASC and the efforts of Bob Linn and crews of the M/Vs *American Spirit*, *American Integrity* and *Burns Harbor*. Thanks are also due to Marc Heatley of Proper Design, Cardiff U.K. for data formatting, and Rich Mueller, NetsCo., for logistical support. This is contribution No. UMCES 5631 from the University of Maryland Center for Environmental Science and contribution No. IMET 19-010 from the University of Maryland Institute for Marine Environmental Technology.

**Conflicts of Interest:** The authors declare no conflicts of interest. ASC had no role in the design of this study or data interpretation. ASC crews were responsible for sample collection as part of their normal ship operations and according to strict procedures laid down by C. L. Mitchelmore (University of Maryland, CES, who supplied the sampling equipment and supervised sample shipment and preparation). Publication and dissemination of results was entirely the decision of the authors.

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
