# Peer review of "Genomic and Microscopic Analysis of Ballast Water in the Great Lakes Region"

_applsci, doi:10.3390/app9122441_

Round 1

Reviewer 1 Report

Review of the paper „A survey of phytoplankton…“ authored by Wright et al.

The authors surveyed the “… phytoplankton community in ballast water aboard great lakes freighters”. This is an interesting and relevant topic, since ballast water is known to be key for the dispersal of aquatic organisms on a global scale leading to numerous invasion events. The authors combined modern genetic techniques with classical light microscopy to study water samples taken during the filling process of the ballast tanks. Although the authors collected many data and information I have a number of concerns, the main concern is that the genetic and microscopy data are not connected.

Major issues:

The abstract provides a lot of background information, but lacks a clear result part. The reader gets no information on the outcome of the comparison of the two methods (see also Results).

The introduction provides a broad overview over the field but it does not specifically focus on the present study. For example, the part on the different ways to treat the water samples (UV-C, NaOH) is too long and might be a subject for the discussion. The present study only investigates untreated water.

The Results are very unbalanced. The microscopic data are presented in almost 40 pages of tables and the genetic data in a number of figures. This needs to be improved. The microscopic data need much more analysis and graphical representation. The tables can be part of an appendix. The main problem with the results section is the lack of comparison genetic and microscopic data. There is only the short part in line 395 stating that OTUs were not assigned to family/genus level. Is this due to the resolution? Then this message is a result and should be presented as such (and in the abstract). However, it should be possible choose a resolution/taxonomic level at which a comparison can be made. The reader cannot judge whether the two methods match or if (and when) discrepancies occur. Without an understandable comparison (with statistical measures) this is not more than a collection of data, though very valuable data.

The discussion

As a consequence, since no clear comparison of the methods is presented, the discussion falls short. If the two methods deliver diverging results, then this divergence would be the result and needs to be discussed. Another general issue. I see the point that it is scientifically interesting to study phytoplankton samples from ballast water using genetic tools, however the practical use i.e. when the freighter is on its way did not become clear to me, since it takes time to prepare the samples and to work on the past-sequencing bioinformatics pipeline.

Please check all species names for typos!

In summary, I think the manuscript needs improvements in scientific issues i.e. clearly formulated hypotheses which are then tested and discussed and also in style i.e. the figures and legends can be much more informative.

Author Response

The points made by this reviewer are well taken, and the manuscript has been extensively revised including a new section specifically comparing the visual (microscopic) and molecular techniques used to characterise the flora contained in the ballast water. We are aware that this approach is speculative, and such a comparison seems to show that it may prove more appropriate for some taxonomic groups more than others. 

The title has been changed to reflect the two different study methods, which have now been subjected to further analysis. This change in scope is also reflected in the addition of another author. The literature has also been updated to include recent papers advocating the use of molecular techniques to characterise biota contained in ballast water. However it is clear from all these papers, including this one, that further refinement is needed to improve the usefulness of this approach in the context of ballast water and studies of aquatic invasive species in general. 

We agree that the draft submitted was difficult to 'digest', and have sought to edit some of the background material as requested, including unnecessary detail on ballast water treatment methods, and (hopefully) corrected the mistakes. We have also added some more recent references.

We agree that the main body of the text was overloaded with data, and most of the tables (those showing microscopic counts) have now been moved to an Appendix. Additionally, several of the figures from the original draft have also been moved to the Appendix. Remaining figures relate to Principal Components Analysis describing the role of season/temperature/location in accounting for qualitative changes in organism densities, and PCA relating to the comparison of visual vs. molecular methodology. The potential for such methodology remains to be seen. In making this comparison we find that visual and molecular techniques can show marked differences depending on the taxonomic group. As an indicative measure of the presence (or removal) of potentially harmful microbiota it would require substantial further refinement. With or without the potential for ballast water compliance assessment, we feel these results as they stand will be of general interest in the study of shifts in microorganism communities according to season, location and possible invasions.     

Reviewer 2 Report

The study entitled “A Survey of Phytoplankton in Ballast Water Aboard Great Lakes Freighters” aims to investigate the diversity of phytoplankton through High Throughput Sequencing. Different scenarios have been presented such as different locations and seasons.

In my opinion, since ballast water topic is a global challenge that covers different scientific fields, the focus of this study can be a bit far from the goals of the special issue “Innovative Water and Wastewater Treatment Technologies for Supporting Global Sustainability” within the Chemistry Section. I think it is a biological/ecological study. Besides, there is a number of weakness that prevent myself from giving this work a favorable assessment.

GENERAL COMMENTS:

1.- Overall, the manuscript needs major improvements in terms of writing and readability. Some sharp, and sometimes confusing sentences appear throughout the manuscript, being difficult to understand. I get lost many times.

Additionally, I have the feeling that the manuscript is carelessly prepared. I found in several occasions typographical erros, e.g. line 43: “has been estimated at >$100 bn. [3] although perhaps…”; line 50: “February 2004. [6]. This”; line 56: “size class >10&<50μM”; line 65: “disinfection biproducts”; line 279-281: “PlamaVolume”; line 325 “by High Throughput Sequencing HTS)”; Line 345: “Using the QUIIME (QIIME?)”; Line 423: “(Elskus et al. [11] demonstrated the need for a seasonal testing regime, there is also a clear requirement for information on the seasonal and temporal variability of the phytoplankton flora in the Great Lakes… ” When is the parenthesis closed?.

2.- There is a misunderstanding in the introduction when authors refer to size class organisms. As I understand, any organisms <50 um, 10-50 um is regulated (because these size classes are in D2 standard). The only unregulated fraction is <10 um, which only is regulated by three specific indicator microorganisms related to human health. Paragraphs such as the line 94-105 must be rephrased, because I have the feeling that authors state that species <50 um are unregulated and it is not correct. Other thing is that some phytoplankton is higher than 50 um in size, but regulations are specific for size class and not for type of microorganism…

3.- Authors must define the goal of the study. I think it is not clearer. Also they must clearly state the conclusions of the study.

3.- Definitively, there is huge amount of tables from page 10 to page 49 that is impossible to interpret it. It has no sense. In its current state the paper needs to be restructured, made more concise and material should be condensed. I would recommend that these format issues be addressed and then the article could be resubmitted for consideration.

4.- The same with Figures. Too much figures with very low information. For example, Fig. 4 and 5 has on the left side other figure that is unintelligible. Fig. 8 cannot be interpreted it. In general, Figures must be concise and material should be condensed. Figure captions are poor and deserve more explanation.

SPECIFIC COMMENTS:

Abstract:

The version online and manuscript version are not equal. This paragraph is missed: “It is estimated that between 3-12 billion tons of ballast water are transported globally each year, resulting in the transfer aquatic organisms to new coastal environments, often with adverse consequences. As much of 80% by volume of international trade is carried by ships, with projected increases in vessel size and speed. Ships’ ballast tanks may hold in excess of 100,000 tons of ballast water, transporting as many as 3000-7000 species every day.”

Line 33-36: Numbers are old: “3-12 billion tons of ballast water”; “3000-7000 species every day”. I encourage the author to refresh the numbers and state current data about ballast water tons and number of species transferred. Also references [1,2].

Line 37: Confusing sentence?: Human health may also be affected both directly (bacterial pathogens) or indirectly (loss of food resources).

Line 43: >$100 “bn??”. although perhaps. “A” Superscript.

Line 57: (IMO 2004). It is a reference, stated as such in the manuscript.

Line 66-68: “The reliance of electrochlorination on the presence of salt in the ballast water together with concern over the release of chlorine into freshwater ports has led to the testing of a sodium hydroxide (NaOH)-based ballast water treatment system aboard Great Lakes freighters in 2015 and 2016” Reference?

Line 78: <10μm (bacteria) size classes. It is not only bacteria…

Line 93-97: >50 um is already in regulations, so it is considered. Please, rephrase.

Line 114: Pfeisteria and Gonyaulax should be cursive?

Methods:

Figure 1 is difficult to see. It seems that it copy-paste from elsewhere. Please, put it accordingly and all text should be as a Figure caption below.

Page 7 and 8. The mix of images for equations are chaotic, please, put it accordingly to the text, on page 7 and table 1.

Results:

Fig. 2: Where is the legend? What does each color mean?

I´m lost, are authors talking about bacteria or phytoplankton? Or both? It must be specified.

PCA Analysis deserves deeper interpretation.

References:

It must be updated, only 7 out of 29 references are from 2015 onwards.

For example, there is a recent review that covers some of the topics that authors discussed in present paper and can help to update the study:

Hess-Erga, O.-K., Moreno-Andrés, J., Enger, Ø., Vadstein, O., 2019. Microorganisms in ballast water: Disinfection, community dynamics, and implications for management. Sci. Total Environ. 657, 704–716. https://doi.org/10.1016/J.SCITOTENV.2018.12.004

Author Response

We have substantially revised this manuscript including the corrections requested by this referee whose comments/criticisms were most helpful. We have also updated the references to include several recent studies advocating the use of molecular techniques to characterise the bacterial and phytoplankton flora found in ballast water. Recently published work describes molecular characterisation of ballast water, the rationale being that there remain serious difficulties in interpreting and applying current, size-based regulations governing ballast water discharge. Following recent studies of ballast water from coastal U.S. ports and major international ports, this study, focussed on freshwater, Great Lakes ports seeks to investigate a molecular approach to characterising the uptake ('challenge') water. While this is still somewhat removed from water treatment per se., there is a recognised potential for such techniques to provide important background information regarding the molecular profiles of potentially harmful invasive species. Establishment of such profiles and their changes following treatment have implications for assessing the efficacy of treatment technologies.Whether there is potential indicative testing for discharge compliance remains to be seen.

As pointed out this study is primarily at the survey stage. However we note the referee's criticism that the initial draft lacked specific visual/microscopic component. We have, therefore added a comparison between microscopic and molecular data.The title has been changed to reflect the two different study methods, which have now been subjected to further analysis.This change in scope is also reflected in a change in title and the addition of another author. As noted above the literature has also been updated to include recent papers advocating the use of molecular techniques to characterise biota contained in ballast water. However it is clear from all these papers, including this one, that further refinement is needed to improve the usefulness of this approach in the context of ballast water and studies of aquatic invasive species in general.

We have corrected the errors/inconsistencies outlined in referee's point 1, paragraph 2 (lines 37, 48, 57,78 under SPECIFIC COMMENTSand have made efforts to clarify confusing sentences. We have deleted reference to disinfection by-products as irrelevant to the primary theme of the paper (Lines 66-68). We have also made an effort to clarify the definition of regulated vs. unregulated size class. The > 50uM size class is of course regulated, although there is an assumption that this size class relates to zooplankton whose allowable discharge density is one millionth of the 10-50uM (phytoplankton) size category. Where large blooms of large (>50uM) dinoflagellates occur, and bearing in mind that movement remains the primary live/dead, diificulties remain in assessing viabilty of this size class if zooplankton are not specified. We have added the review by Hess-Erga et al. (thanks for this reference) who, among others, stress the omission of the <10uM size class.   

We agree that the main body of the text was overloaded with data, and most of the tables (those showing microscopic counts) have now been moved to an Appendix as Appendix 1, Tables 1-7. Additionally, several of the figures from the original draft have also been moved to the Appendix. Figures remaining in the body of the text relate to Principal Components Analysis describing the role of season/temperature/location in accounting for qualitative changes in organism densities, and PCA relating to the comparison of visual vs. molecular methodology  

Round 2

Reviewer 1 Report

Review of the revised version of manuscript: A Molecular and Microscopic Survey of Ballast Water Aboard Great Lakes Freighters. authored by Wright et al.

I think the manuscript has improved considerably, but I must say, I am still not happy with the current version.

My major concern is the rationale of the manuscript. What has been done basically was the sampling of water in a harbour using the the technique of filling ballast tanks of the ships. The untreated water was microscopically and genetically (genomically) analysed. The result according to Fig. 6 was that the two methods deliver very differing results demonstrated by the almost exclusive separation in the PCA (please present axis loadings for the two axes). For me the key message is that HTS is not an adequate methods for analysing phytoplankton samples in general (and therefore also not adequate for balast water). Since the water in the present study remained untreated and no samples were taken after the cruises, the story around the ballast tank dispersal is far-fetched; just harbour water was analysed in two different ways.

Another issue: even a decent match of the two methods would not be very helpful for combatting dispersal of phytoplankton and bacteria, because the HTS is a time-consuming procedure and will deliver data only after the cruises of the freighters. 

For me the manuscript is a genomic and microscopic analysis of phytoplankton and baceria from harbours in the Great Lakes region.

Author Response

We agree with this referee that the suggestion that HTS may qualify as a potential indicative tool for ballast water management represents an over-reach, particularly in view of our most recent data analyses. We have, therefore, significantly modified the introduction and discussion to de-emphasize HTS as a candidate for prognostic testing. Certainly the discrepancies between Molecular vs. microscopic assignment argue against its adoption as an indicative test and this point has been made in the text. Additionally the observation that, currently, HTS analysis takes too long to fulfil the time requirements for sampling/testing during a port visit has been added to the discussion. Both the Introduction and the discussion have been edited to de-emphasize the indicative potential. However we believe that the results presented will be of potential useful in elucidating the presence and transport of invasive species. As such, it is hoped that these data can contribute to a risk-based approach to this issue. Like several recent molecular surveys there is good potential for source water to be identified, along with possibilities for identifying potentially harmful populations of microorganisms. While HTS has not been applied to treated water in this or other recent surveys, this remains an option, although potential for direct ballast water management remains speculative  

As stated by this referee, the current paper is really a survey. Accordingly we have altered the title along suggested guidelines, although as with similar published surveys, we have kept the words 'Ballast Water' to enable people interested in this topic to navigate via a word search.

The comments of this referee are much appreciated, and hopefully have resulted in a clearer and better reasoned rationale   

Reviewer 2 Report

The study entitled “A Molecular and Microscopic Survey of Ballast Water Aboard Great Lakes Freighters” has improved after revisions made by the authors.

I am still thinking that since ballast water topic is a global challenge that covers different scientific fields, the focus of this study can be a bit far from the goals of the special issue “Innovative Water and Wastewater Treatment Technologies for Supporting Global Sustainability” within the Chemistry Section.

GENERAL COMMENTS:

Authors must clearly define the goal of the study and clearly state the conclusions based on the results that have been explained throughout the manuscript.

The Authors should ameliorate the quality of the figures: they are grainy and sometimes unclear as in the case of the legend on Figure 4 and 5.

I think that some improvements can be made in order to increase the quality of the manuscript. Mainly in methods and results section. They are "hard" to read and interpret it. 

SPECIFIC COMMENTS:

Abstract:

It is written in a proper way. Typo error on line 26: “..”

Introduction:

It is quite good introduction that update the ballast water topic problem with studies that uses real data in different geographical areas.

Line 56-58. I found confusing too much symbols related to the size class. Maybe is better to combine it in a table. CFU should be in superscript.

Methods:

Equations on page 6 and 7 should be written as equations and not as a Figure.

In my opinion, Table 1 is not necessary. Besides, it is not referenced in the text. Maybe in supplementary material.

Results:

Revise the species name on lines 308-309: Aphamiziminon?

Discussion:

Authors state in the introduction some concerns about organisms <10 um in size. The discussion would be enriched if authors state a paragraph by summarizing and discussing his results under this topic.

CONCLUSIONS??

Author Response

We agree with this referee that the stated rationale behind this project and aspects of the conclusions and discussion require clarification. Hopefully we have done so in this draft. The project was one of several recent HTS surveys of ballast water, with some initial expectation that results would provide useful information on the presence/transport of invasive species, and perhaps some potential for prognostic testing. The latter goal seems more speculative than we had initially anticipated, although potential for a risk-based approach based on invasive species sources and appearances remains positive.

As the project has evolved the application of HTS as a universal indicative test  has proved less attractive as a tool for ballast water management, although it has yet to be tested on treated water. As suggested by this referee, a section has been added to the Discussion emphasizing the need for analytical techniques that are independent of the (confusing) regulated size categories. We have edited the Introduction and Discussion to de-emphasize the prospect for HTS in ballast water management, particularly in light of the poor correlation between molecular and microscopic assignment. Also HTS is currently too time consuming to be accommodated with a ship's port visit. Our edits reflect these changes and have been designed to better define the stated rationale and objectives. Conclusions on the limited scope of HTS for ballast water management are incorporated in the Discussion

Line 56-58 (Introduction). Symbols relating to regulatory categories/size classes have been removed for clarification (CFUs spelt out in full)

 The Methods section has been edited with the removal of Figure 1, as suggested.

Aphamizinenon (L. 308/9) has been corrected.

Equations on pages 6 & 7 are now as equations, not figures.

To clarify some of the figures we would support the action of their inclusion in electronic format via links if possible.

Round 3

Reviewer 2 Report

Last version improves significantly from the first one. I think the study now is well described and results supports the conclusions. 

Other thing is the goal of the study, which, under my particular opinion, I think it is a bit far from objectives of an special issue called “Innovative Water and Wastewater Treatment Technologies for Supporting Global Sustainability” because all results provided coming from untreated samples. 

Author Response

The comments from this referee are well taken, and we accept that the justification of this study within the context of "Innovative Techniques for Water Treatment Technologies" requires a broadening of the scope into Risk Assessment, although the use of HTS as a specific indicative test (of ballast water) remains speculative. The conclusions reflect a (justifiably) less optimistic, or at least more limited view of this application, and we suggest that 'negative' results can be just as instructive as 'positive' data. Nevertheless we hope to expand this line of investigation to HTS analysis of treated water. Meanwhile the submission is made as an extension of this technology into the freshwater environment, with potential (after further refinement) as a risk assessment (and perhaps indicative) tool for ballast water or invasive species in general.